

# Glacier variations in the Himalaya from 1990 to 2015 based on remote sensing

Qin Ji[1,2], Jun Dong[3], Hong-rong Li[4], Yan Qin[1], Rui Liu[2], Tai-bao Yang[1*],

[1]Institute of Glaciology and Ecogeography, College of Earth and Environmental Sciences, Lanzhou University, Lanzhou, China

[2]GIS Application Research Chongqing Key Laboratory, School of Geography and Tourism, Chongqing Normal University, Chongqing, China

[3]Chongqing Business Vocational College, Chongqing, China

[4]PLA Army Special Operations Academy, Guangzhou,China

**ABSTRACT.** The Himalaya is located in the southwest margin of the Tibetan Plateau. The region is of special interest for glacio-climatological research as it is influenced by both the continental climate of Central Asia and The Indian Monsoon system. Despite its large area covered by glaciers, detail glacier inventory data are not yet available for the entire Himalaya. The study presents spatial patterns in glacier area in the entire Himalaya are multiple spatial scales. We combined Landsat TM/ETM+/OLI from 1990 to 2015 and ASTER GEDM (30 m). In the years around 1990 the whole mountain range contained about 12211 glaciers covering an area of 23229.27 $km^2$, while the ice on south slope covered 14451.25 $km^2$. Glaciers are mainly distributed in the western of the Himalaya with an area of 11551.69 $km^2$ and the minimum is the eastern. The elevation of glacier mainly distributed at 4,800~6,200 m a.s.l. with an area percent of approximately 84% in 1990. The largest number and ice cover of glaciers is hanging glacier and valley glacier, respectively. The number of debris-covered glaciers is relatively small, whereas covers an area of about 44.21% in 1990. The glacier decreased by 10.99% and this recession has accelerated from 1990 to 2015. The average annual shrinkage rate of the glaciers on the north slope ($0.54\% \ a^{-1}$) is greater than that on the south slope ($0.38\% \ a^{-1}$). Glacier decreased in the debris-covered glaciers and debris-free glaciers, and the area loss for the first is about 15.56% and 5.22% for the latter during 1990-2015, which showed that the moraine in the Himalaya can inhibit the ablation of glaciers to some extent.

**Key words:** Himalaya, glacier variations, climate change, remote sensing

## 1. Introduction

Cryosphere refers to the negative temperature layer with continuous and a certain thickness on the surface of the earth, including glaciers, ice caps, ice sheets, snow, permafrost, and river and lake ice (e.g. Qin et al., 2009; Bolch et al., 2019). As a major component of the cryosphere, glaciers play an important role in climate system, which are widely recognized as a key indicator for early detection for the impacts of global climate variations in remote regions where the weather station are rarely (e.g. Masiokas et al., 2008; Yao et al., 2012). Glaciers store 70% of global freshwater resources and are regarded as a natural solid reservoirs having a great regulation effect on river runoff, especially for the arid and semi-arid areas in the middle and low latitude mountainous regions, which can collect solid precipitation in winter and release it with a seasonal delay in the form of meltwater, just when it is needed most urgently for agriculture and as drinking water (e.g. Kaser et al., 2010; Xie and Liu,



2010), thus reducing the impact of annual runoff changes (Röthlisberger and Lang,
1987). Although glacier variations can provide a large amount of water resources of
downstream populations and bring economic benefits to society, the melting and
movement of glaciers will also cause many natural disasters, such as glacial lake
outbursts flood (GLOFs) and sea-level rise (e.g. Meier et al., 2007; Bajracharya and
Mool, 2009). Although the accelerated retreat of ice sheet contributes a lot to sea-level
rise, the effect of a large number of meltwater from the retreat of mountain glaciers
should not be underestimated (e.g. Arendt et al., 2002; Jacob et al., 2012; Marzeion et
al., 2015, 2017; Richter et al., 2017). Dyurgerov and Meier (1997) investigated the
mass balances of all small glaciers in the world (except the Antarctic and Greenland
ice sheets) to estimate their annual variation and determine their contribution to the
changes of sea level, which found a new global mass balance value, averaging -130 $\pm$
33 mm yr$^{-1}$, totaling -3.9 m in water equivalent for 1961-1990, or 0.25 $\pm$0.10 mm yr$^{-1}$
in sea-level equivalent. This is about 14 to 18% of the average rate of sea-level rise in
the last 100 yr. Garnder et al. (2013) estimated the global mass budget was $-259 \pm 28$
gigatons per year, equivalent to the combined loss from both ice sheets and
accounting for 29 $\pm$13% of the observed sea level rise between 2003 and 2009.
The Himalaya is located in the southwestern margin of the Tibetan Plateau and
regarded as "geographically critical areas" together with the Alaska and Patagonia
Plateaus, where modern glaciers are dense (e.g. Haeberli, 1998; Meier and Dyurgerov,
2002). Most of glaciers in this region are classified as maritime or temperate-type and
very sensitive to climate change, which is the source area of many major rivers (e.g.
Ganges, Indus, Yangtze, Brahmaputra) (Immerzeel et al., 2010). There is about 800
million people around the world depend on these rivers to survive (Kaser et al., 2010).
Over the past three decades, most of the Himalaya's glaciers have shown a tendency
to shrink (e.g. Bolch et al., 2008; Yao et al., 2012). In addition, shortages and
utilization of freshwater recourses in the Himalaya may also lead to international
disputes as an important strategic national resource (Zhang et al., 2009), and the
shrinkage of glaciers will also result in sea-level rise, which will flood large areas
along the coast. Therefore, the distribution and changes of glaciers in the Himalaya
have always attracted the attention of the scientific community (e.g. Ma et al., 2010;
Bhambri et al., 2011; Li et al., 2011). Moreover, the formation and evolution of the
Himalaya are important to the atmospheric circulation and climate change in Asia and
the world, and the study of glaciers and environmental changes in the Himalaya has
important scientific significance (Shi et al., 2005).
Previous studies about glacier distribution and changes in the Himalaya have
focused mainly on individual glaciers or river basins (e.g. Ye et al., 2007; Nie et al.,
2010; Yin et al., 2012; Liu et al., 2013; Bolch et al., 2012; Yao et al., 2012; Immerzeel
et al., 2014). The glacial distribute area more and the types are diverse, and the terrain
and climatic conditions are also complex in this region. Therefore, the scale of
individual glaciers or river basins is not enough to reflect the changes of glacier
throughout the Himalaya. In this paper, we selected the entire Himalaya as the
research region used remote sensing and GIS technology to analyze the glacier



distribution and variation characteristics in the past 25 years. Therefore, the aims of this study are: (1) to generate glacier extents for the Himalaya from 1990 to 2015, (2) to provide information on the characteristics of glacier distribution and (3) to analyze the dynamics of glacier changes in different regions, elements and forms.

**2. Study area**

The Himalaya Range, situated in the border of China, Pakistan, India, Nepal and Bhutan (Fig. 1), is the highest mountain in the world with the highest peaks at ~8844.43 m a.s.l where snow covered throughout the year (Fig. 1b), and the main part is located at the international boundary between Nepal and China where the glacial meltwater through the Indus, Ganges, Yarlung Tsangpo-Brahmaputra and eventually drained into the Indian Ocean (Shi et al., 2005). The Himalaya can be divided into three sections (Fig. 1a) (Qin, 1999). The western Himalaya is under the complex influence of both the continental climate and the Indian Monsoon system with the westerlies in winter and Indian monsoon in summer (e.g. Bookhagen and Burbank, 2006; Krishna, 2018); The east Himalaya is closed to the Yarlung Tsangpo valley where warm, wet monsoonal air masses cross the area predominantly in summer and transport abundant precipitation, with cumulative precipitation of 1,000–3,000 mm, the highest average precipitation rate of the entire Tibetan Plateau (Yang et al., 2008).

As the division between the water cycle and climate, the Himalaya plays a decisive role in the meteorological conditions between the Indian subcontinent in the southern and the Central Asian highlands in the northern. The southern slope faces the Indian monsoon with abundant precipitation and the largest precipitation zone generally appears at 2,000 m a.s.l. Compared with the southern slope, the Himalaya, especially the Greater Himalaya, blocks off the cold air mass from the northern part into India in winter, and on the other hand forces the southwest monsoon to give up a lot of moisture before moving northward through the mountains. Thus, the precipitation on the northern slope is significantly reduced, such as the annual precipitation of about 335.1 mm in 1959 recorded at the Rongbu temple weather station (northern slope; 5,000 m a.s.l), and the lower altitudes reduced to 236.2 mm observed at the Dingri weather station (northern slope; 4,300 m a.s.l) (Li et al., 1986).

Glaciers on the Himalaya are roughly classified into continental and temperature glaciers (Huang 1990). Continental type glaciers are widely distributed from the northern slopes of the western Himalaya to central Himalaya with little precipitation and cold ice, while the maritime type is the eastern Himalaya and southern slope with abundant precipitation and a temperate ice body.

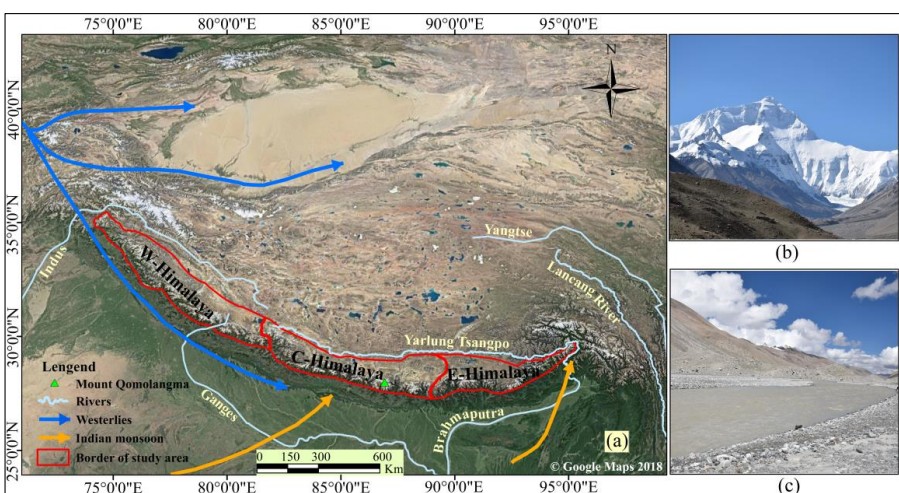

**Fig. 1.** (a) Location of the study area s are overlaid on the Google Earth image; (b) Overview of
the Mount Qomolangma; (c) Glacial melt in the Mount Qomolangma

## 3. Data and methods

*3.1. Data*

The main source for the glacier outlines was Landsat TM/ETM+/OLI scenes from
different years. The scenes were available from USGS (United States Geological
Survey, https://earthexplorer.usgs.gov/) and orthorectified automatically using the
SRTM3 DEM (Level 1T) (Bolch et al., 2010). Guo et al. (2012) demonstrated that
orthorectified Landsat data had high precision, most of them have correction accuracy
of about half a pixel. Clouds, seasonal snow cover, shadows and debris are the major
sources of misclassified areas (e.g. Bhambri et al., 2012; Shangguan et al., 2014). In
order to improve the accuracy of glacier outlines, we selected imagery taken during
the melting season, when the glaciers are less affected by seasonal snow and
additional scenes from similar time periods (about 2 years) were used as alternatives,
which can eliminate to the effects of seasonal snow and clouds in a certain extent.
Besides, the images of different time periods may also have different solar elevation
angles during the acquisition process that is largely conducive to reduce impact of
shadows. There are 200 scenes were used eventually (Table 1).

A DEM of appropriate quality and resolution is required to derive topographic
parameters such as minimum, maximum, and mean elevation, slope, and aspect (Frey
et al., 2012). In view of the free availability of digital elevation models from the
Shuttle Radar Topography Mission (SRTM) from 2000 at about 90 m resolution and
the new ASTER global DEM (GDEM) have high scientific research significance and
value. The SRTM 3 was compiled using interferometry synthetic aperture radar
(InSAR), which is easily affected by specular reflection, echo lag and radar shadow
resulted in missing areas and outliers (Kang and Feng, 2011). The ASTER GDEM
was acquired by setero-image pair of optical imaging, and has a spatial resolution of
30 m.



In a study for western Japan Hayakawa et al. (2008) found, that over glaciers, the
ASTER GDEM is slightly superior to the SRTM 3, particularly in steep terrain, but
both of them can be used to extract glacier inventories. We resampled the ASTER
GDEM to 90 m in our study area and subtract with the SRTM 3 revealed in many
regions differences ranges from -50 to 50 m, which is about 70% (Fig. 2). In addition,
we made the hillshade in parts of the western Himalaya using the DEMs (Fig. 3) and
found that the interpolated terrain in the SRTM 3 is continuous and looks realistic, but
all the interpolated regions are systematically too low, resulting in distinct shadows in
the hillshade view at the margins of these crater-like depressions, ie null areas. We
thus used the ASTER GDEM for this study.
**Table 1** Utilized Landsat scenes

| Path/Row | ~1990 | | | | ~2000 | | | |
| | Acquisition data | Sensor | Cloud cover (%) | Reference data | Acquisition data | Sensor | Cloud cover (%) | Reference Data |
|---|---|---|---|---|---|---|---|---|
| 149/36 | 1990-08-07 | TM | 44 | – | 2001-08-29 | ETM+ | 17 | 1998-08-29 |
| 148/36 | 1991-09-20 | TM | 53 | – | 2000-08-27 | TM | 10 | 1999-08-17 |
| 148/37 | 1991-09-20 | TM | 34 | 1992-10-24 | 2000-09-04 | ETM+ | 23 | 1999-08-17 |
| 147/37 | 1991-08-28 | TM | 64 | 1992-08-14 | 2000-08-28 | ETM+ | 24 | 2001-09-08 |
| 147/38 | 1989-08-06 | TM | 25 | 1992-08-14 | 2001-09-24 | TM | 1 | 2002-08-02 |
| 146/38 | 1992-11-11 | TM | 1 | – | 2001-09-09 | ETM+ | 2 | 2001-08-24 |
| 146/39 | 1992-11-11 | TM | 8 | – | 2000-08-05 | ETM+ | 13 | 2000-10-08 |
| 145/38 | 1990-11-15 | TM | 3 | – | 2000-11-02 | ETM+ | 2 | 2000-09-15 |
| 145/39 | 1990-11-15 | TM | 1 | – | 2001-08-01 | ETM+ | 26 | 2001-10-20 |
| 144/39 | 1991-12-13 | TM | 42 | 1992-11-13 | 1999-12-03 | TM | 17 | 1998-10-13 |
| 143/39 | 1988-12-13 | TM | 3 | 1991-12-06 | 2000-10-03 | ETM+ | 1 | 1998-09-04 |
| 143/40 | 1988-10-26 | TM | 13 | 1991-12-06 | 2001-12-09 | ETM+ | 16 | – |
| 142/40 | 1991-10-12 | TM | 0 | 1988-10-19 | 2000-12-15 | ETM+ | 2 | 2001-10-31 |
| 141/40 | 1988-12-15 | TM | 2 | 1992-09-21 | 2000-11-22 | ETM+ | 1 | 2001-09-22 |
| 140/40 | 1989-11-09 | TM | 1 | 1992-11-17 | 2000-11-15 | TM | 0 | 2000-12-09 |
| 140/41 | 1989-11-09 | TM | 1 | 1990-08-24 | 1999-04-27 | TM | 27 | 2000-10-30 |
| 139/40 | 1990-06-14 | TM | 0 | 1988-06-08 | 2001-12-29 | ETM+ | 1 | – |
| 139/41 | 1990-06-14 | TM | 42 | 1998-12-01 | 2000-12-26 | ETM+ | 1 | 2000-11-08 |
| 138/40 | 1990-01-14 | TM | 1 | 1991-11-01 | 2000-12-19 | ETM+ | 1 | 1998-11-04 |
| 138/41 | 1991-10-16 | TM | 24 | – | 1999-09-20 | TM | 32 | 1998-11-04 |
| 137/40 | 1988-09-30 | TM | 24 | 1991-10-09 | 1999-05-08 | TM | 0 | 2000-12-28 |
| 137/41 | 1988-09-30 | TM | 16 | 1988-09-14 | 2000-12-28 | ETM+ | 0 | 2000-10-17 |
| 136/40 | 1988-10-09 | TM | 24 | 1989-06-22 | 1998-12-08 | TM | 11 | – |
| 136/41 | 1990-06-25 | TM | 24 | 1988-10-09 | 2001-01-30 | TM | 21 | 1998-12-08 |






**Table 1** (continued) Utilized Landsat scenes

| | ~2010 | | | | ~2015 | | | |
|---|---|---|---|---|---|---|---|---|
| Path/Row | Acquisition data | Sensor | Cloud cover (%) | Reference data | Acquisition data | Sensor | Cloud cover (%) | Reference data |
| 149/36 | 2008-07-31 | ETM+ | 15 | 2009-08-27 | 2016-10-01 | OLI/TIRS | 1 | 2015-09-13 |
| 148/36 | 2008-08-25 | ETM+ | 12 | 2008-07-24 | 2016-10-02 | ETM+ | 1 | 2015-08-29 |
| 148/37 | 2008-08-25 | ETM+ | 23 | 2008-07-24 | 2015-09-14 | ETM+ | 29 | 2016-09-08 |
| 147/37 | 2008-08-25 | ETM+ | 51 | 2009-08-29 | 2016-10-03 | OLI/TIRS | 1 | 2015-09-15 |
| 147/38 | 2011-09-28 | ETM+ | 25 | 2010-09-09 | 2015-09-15 | OLI/TIRS | 7 | 2015-08-30 |
| 146/38 | 2008-07-28 | ETM+ | 18 | 2011-09-13 | 2015-09-16 | ETM+ | 2 | 2015-09-08 |
| 146/39 | 2011-12-26 | ETM+ | 0 | 2012-09-23 | 2016-11-13 | OLI/TIRS | 10 | 2016-11-29 |
| 145/38 | 2009-07-30 | TM | 21 | 2011-09-22 | 2015-09-17 | OLI/TIRS | 3 | 2015-10-03 |
| 145/39 | 2011-09-22 | TM | 30 | 2009-07-30 | 2016-12-08 | OLI/TIRS | 1 | 2016-11-06 |
| 144/39 | 2011-10-09 | ETM+ | 2 | 2011-10-01 | 2015-09-10 | OLI/TIRS | 3 | 2015-09-26 |
| 143/39 | 2011-10-18 | ETM+ | 2 | 2009-07-24 | 2015-09-03 | OLI/TIRS | 3 | 2015-10-05 |
| 143/40 | 2011-12-05 | ETM+ | 37 | 2011-11-19 | 2015-09-27 | ETM+ | 32 | 2015-10-05 |
| 142/40 | 2009-09-27 | TM | 23 | 2012-10-13 | 2015-10-06 | ETM+ | 1 | 2015-09-28 |
| 141/40 | 2010-12-12 | TM | 24 | 2010-06-19 | 2015-10-07 | OLI/TIRS | 3 | – |
| 140/40 | 2009-11-08 | ETM+ | 1 | 2008-09-02 | 2015-09-30 | OLI/TIRS | 3 | – |
| 140/41 | 2012-12-02 | ETM+ | 1 | 2012-06-09 | 2015-10-08 | ETM+ | 1 | 2013-01-03 |
| 139/40 | 2010-04-18 | TM | 8 | – | 2015-10-09 | OLI/TIRS | 1 | – |
| 139/41 | 2012-10-08 | ETM+ | 18 | 2012-12-11 | 2015-10-09 | OLI/TIRS | 17 | 2013-10-11 |
| 138/40 | 2009-01-04 | TM | 1 | 2011-09-05 | 2015-09-08 | ETM+ | 11 | 2015-10-02 |
| 138/41 | 2008-01-16 | TM | 13 | 2008-10-22 | 2015-09-08 | ETM+ | 55 | 2015-10-26 |
| 137/40 | 2009-11-11 | TM | 0 | 2012-12-29 | 2015-09-09 | OLI/TIRS | 2 | 2015-09-25 |
| 137/41 | 2011-09-30 | TM | 8 | 2009-12-13 | 2016-09-09 | OLI/TIRS | 36 | 2015-10-27 |
| 136/40 | 2009-11-04 | TM | 14 | 2011-08-30 | 2013-09-28 | OLI/TIRS | 6 | – |
| 136/41 | 2009-11-04 | TM | 5 | 2008-10-15 | 2015-11-21 | OLI/TIRS | 1 | 2013-09-28 |

**Note: "–"** represents no imagery as reference

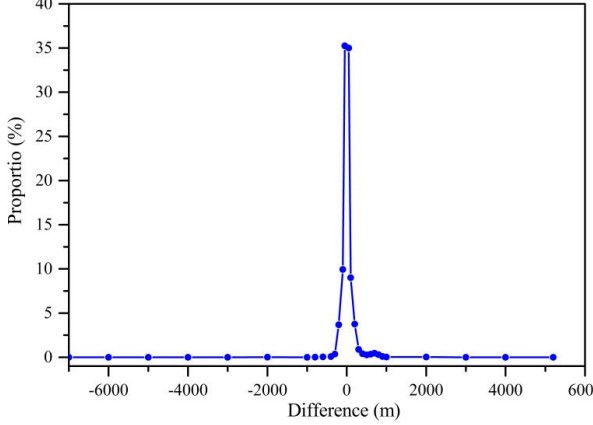


**Fig. 2.** Difference between the ASTER GDEM and the SRTM 3





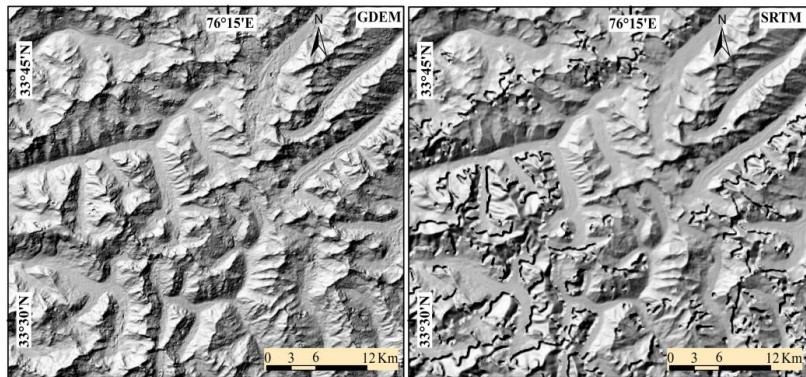


**Fig. 3.** the hillshade view of the ASTER GDEM and the SRTM 3

*3.2. Methods*

*3.2.1. Mapping of glacier*

Compared to other methods of extracting glacier borderlines, segmentation of ratio is considered a robust and convenient algorithm, which is based on the fact that ice has a high reflectivity in visible spectrum and a low reflectivity in shortwave infrared spectrum (Sidjak And Wheat, 1999; Paul et al., 2002; Andreassen et al., 2008). Previous study indicated that B3/B5 is better than B4/B5 to extract glacier extents, which is marked by shadows and debris-cover (Bolch et al., 2010). We also used the semi-automated method to extract glacier outlines as the follow steps, (1) created the ratio image, which was B3/B5 for the Landsat TM and ETM+ imagery and B3/B6 for the Landsat OLI imagery, (2) determined the threshold. After creating the ratio image, we selected 1.8 and 1.0 to produce glacier outlines, respectively, (3) created the binary image. A ratio greater than or equal to the threshold could be assigned 1 and identified as a glacier, and (4) converted these grid data to vector data. To eliminate features that were most likely snow patches or isolated pixels, a 3 by 3 median filter was applied. We visually checked glacier polygons derived from the ratio approach. For debris-free glacier, seasonal snow is the main influencing factor. In the main process of visual interpretation, we referred to the Second Chinese glacier inventory for comprehensive identification. The termini of some debris-covered glaciers were difficult to automatically identify by the ratio method because the spectral characteristics of the debris-covered parts are similar to those of the surrounding surface (Fig. 4), and the more time-consuming part of the glacier mapping was required in the post-processing stage. Paul et al. (2002) though that ice crevasse and debris-covered ice connected to the main glaciers should be considered a part of the glaciers, while seasonal snow, dead ice and ice lakes are not belong to the glaciers. Here, we used several rules to identify the most likely position of the termini: (1) if there is supraglacial ponds or ice cliffs, the end of the glaciers can be determined according to the location of the supraglacial ponds or the cast shadow of the ice cliff (Fig. 5a), (2) if there are creeks in the flat area at the end of the terminus, the glacier boundary can be determined based on the location of the creeks (Fig. 5b), (3) comparing the remote sensing image





in different periods, if the latter images appeared a large number of small lakes and
we can considered it as the debris-covered parts (Fig. 5c and 5d) and (4) combing
Google Earth to distinguish the differences between the color of the glacial terminal
and the surrounding surface. If the color of the glacial terminal is deeper than that of
the surrounding, the region is considered to be debris-covered glacier. The main
reason is that the lower part of debris-cover ice is ice layer with a high water content.
Therefore, the color of debris-covered glacier is deeper than the surrounding surface.

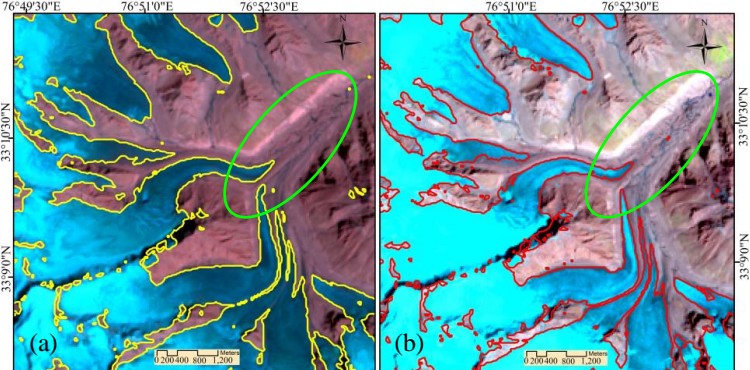


**Fig. 4.** The glacier boundaries by the band ratio (ellipse is the debris-covered glacier). (a) Landsat
ETM+ image and (b) Landsat OLI scene from USGS

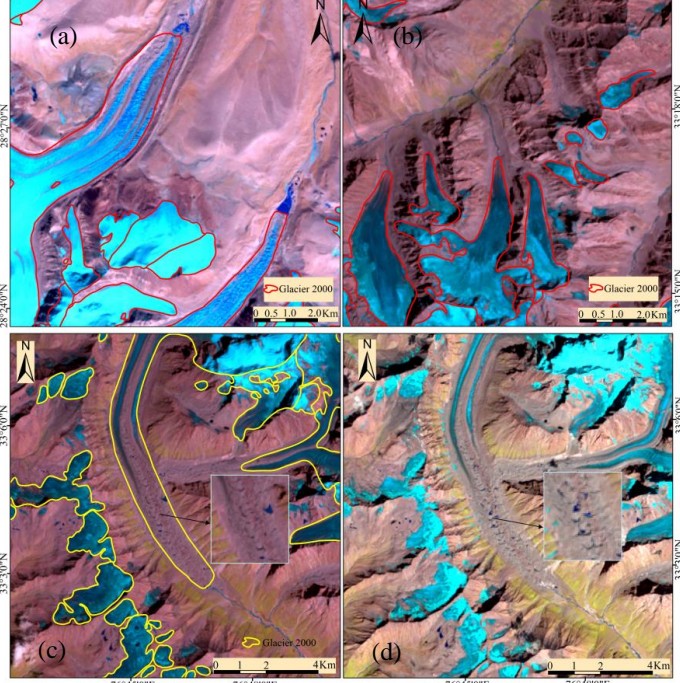


**Fig. 5.** The glacier outlines of debris-covered. (a) supraglacial ponds in the end; (b) creeks in the
end; (c) Landsat ETM+ image in 2000 and (d) Landsat OLI sence in 2015 acquired from USGS





### 3.2.2. Error estimation

Although visual checks were used to correct potential error, there are also some
uncertainties in glacier mapping. Several methods can be used to assess misclassified
areas: (1) field measurements, which has higher accuracy but it is very
time-consuming and labor-intensive so it is generally suitable for small-scale research
(Shangguan, 2007), and (2) multi-temporal uncertainty measurements (e.g. Hall et al.,
2003; Silverio and Jaquet, 2005). To verify the accuracy of the extraction of glacial
boundaries, we compared GPS data obtained in the field with the position of the
terminus of the Zhongni Glacier (debris-covered glacier) and the 5Z342B0021 glacier
(debris-free ice) near the Namurani Peak in the Himalaya, respectively. The results of
the GPS measurement and the visual interpretation in 2015 and 2016 were shown in
Fig. 6. The average distance and standard deviation between the glacier boundaries
and the sampling point as shown in Table 2 and these uncertainties are within the
range of accuracy estimates. Although we used field surveys to validate the results, it
was limited to several glaciers. In order to understand the characteristics of glacier
area changes in more detail, we use the buffer method (15 m) (Bolch et al., 2010) to
calculate the accuracy.

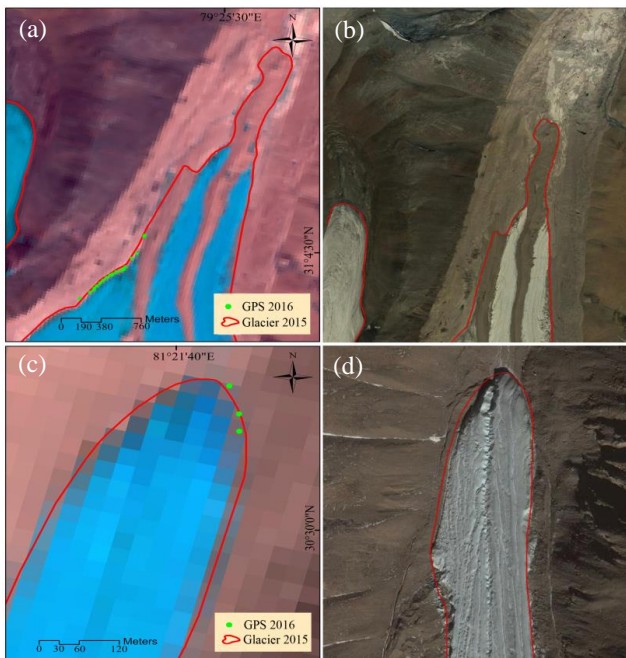


**Fig. 6.** (a) and (b) are the outline of Zhongni Glacier and positions measured using GPS in
Landsat OLI and Google Earth, respectively; (c) and (d) are the boundary of 5Z342B0021Glacier
and positions measured using GPS in Landsat OLI and Google Earth, respectively





**Table 2.** The comparison of Zhongni glacier and 5Z342B0021 glacier between visual
interpretation and GPS measurement

| Name | Acquisition data | Average distance (m) | Standard deviation (m) |
|------|------------------|----------------------|------------------------|
| Zhongni Glacier | 2016-09-19 | 19.6 | 8.9 |
| 5Z342B0021Glacier | 2015-09-26 | 5.7 | 3.5 |

**4. Results and Discussions**
*4.1. Glacier characteristics and change analysis*
*4.1.1. Glacier characteristics and recession for the whole Himalaya*
According to our inventory, glaciers of the whole Himalaya cover an area about
23229.27 km$^2$ in 1990 (Table 3). Ice cover area decreased significantly, with a total
area loss of 2553.10 km$^2$ during the period 1990–2015, equivalent to 10.99% of the
original area in 1990. Shrinkage of the glaciers was about 891.02 km$^2$ (~ –0.38 % a$^{-1}$)
from 1990 to 2000. Percentage loss and rate were in a similar range for the periods
1990–2000 and 2000–2010 but slightly higher for the latter. Glacier area loss of
761.97 km$^2$ with the annual percentage of area retreat about 0.71% a$^{-1}$ in 2010–2015,
was higher than the first two periods. Glaciers shrinkage has accelerated in the
Himalaya over the past 25 years, especially in 2010–2015 (Fig. 7). This is consistent
with the most parts of the Tibetan Plateau.
**Table 3.** Glacier area distribution and change in the Himalaya for 1990–2015

| Year | Area (km$^2$) | Variation (km$^2$) | Variation rate (%) | APAC (% a$^{-1}$) |
|------|------------|-----------------|------------------|----------------|
| 1990 | 23229.27 ± 997.28 | – | – | – |
| 2000 | 22338.25 ± 981.83 | -891.02 ± 15.45 | -3.84 ± 0.07 | -0.38 ± 0.007 |
| 2010 | 21438.14 ± 959.61 | -900.11 ± 22.22 | -4.03 ± 0.10 | -0.40 ± 0.010 |
| 2015 | 20676.17 ± 944.28 | -761.97 ± 15.33 | -3.55 ± 0.07 | -0.71 ± 0.007 |
| Total | – | -2553.10 ± 53.00 | -10.99 ± 0.23 | -0.44 ± 0.014 |

**Note:** APAC is the annual percentage of area change

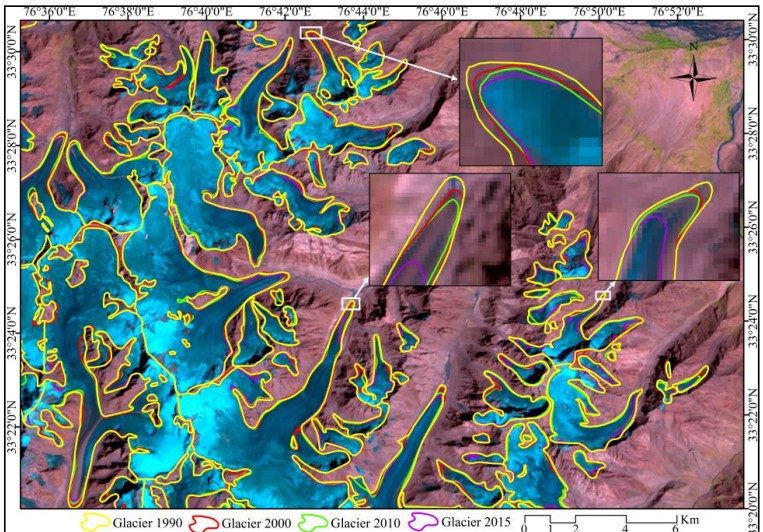

**Fig. 7.** A part of glacier changes during 1990-2015 (Background is Landsat ETM+ 2000/08/28)




In order to understand the glacier distribution characteristics in the Himalaya, we
compared the available recent estimates of glacier area for the entire or regional
Himalaya (Table 4).
**Table 4.** Recent estimates of glacier area for the entire or regional Himalaya

| Study area | Area (km$^2$) | References | Differences with our research (%) |
|---|---|---|---|
| the entire Himalaya | 21,973 | Cogley (2011) | 2.5 |
| the entire Himalaya | 22,829 | Bolch et al. (2012) | 6.5 |
| the entire Himalaya | 19,991 | Nuimura et al. (2015) | 3.3 |
| the regional Himalaya | 4,190 | Guo et al. (2015) | 1.3 |

*4.1.2. Glacier distribution and changes on the north and south slopes*
The south slope of the Himalaya is steep and abundant in precipitation. However,
the north slope is gentle and dry. How to change of the glaciers in the south and north
of the Himalaya under the background of global warming? To answer this question,
we subdivided into two sections in the Himalaya based on the main ridgeline and
analyzed the distribution and changes characteristics. The results are as shown in
Table 5 and Table 6.
Compared the results of the glacier area on the south slope of the Himalaya from
1990 to 2015, it is known that overall glacierized area was about 14451.25 km$^2$ in
1990, and it had been reduced to 13082.14 km$^2$ in 2015, and the number was 5650,
5745, 5816 and 5875, with the average scale of about 2.56 km$^2$ and 2.43 km$^2$, 2.33
km$^2$ and 2.23 km$^2$, respectively (Table 5). The area shrank significantly, with a total
area loss about 1369.11 km$^2$, equivalent to 28.3% of the original area in 1990. The
APAC was 0.38% a$^{-1}$, and the shrinkage rates in different time periods are inconsistent.
The glacier area reduced by 3.30% and APAC was 0.33 % a$^{-1}$ during the period 1990–
2000. In the second period (2000-2010), the glacier area retreated by 431.79 km$^2$,
with APAC about 0.31% a$^{-1}$, which is less than the first period. For 2010–2015, the
annual shrinkage rate of the glacier area was faster than in other intervals. In summary,
the annual retreat rate of the glacier on the south slope has decreased first and then
increased over the past 25 years. Analysis of the average size of the glacier on the
southern slope showed that it has gradually decreased during the period 1990–2015.
The reduction in the average size is likely to be the shrinking of the glacier area and
the increase in the number of glacier.
**Table 5.** Glacier area distribution and changes of southern in the Himalaya during 1990–2015

| Year | Area (km$^2$) | Number | Average size (km$^2$) | Variation (km$^2$) | Variation rate (%) | APAC (% a$^{-1}$) |
|---|---|---|---|---|---|---|
| 1990 | 14451.25 ± 583.40 | 5650 | 2.56 | – | – | – |
| 2000 | 13973.83 ± 572.00 | 5745 | 2.43 | -477.42 ± 11.40 | -3.30 ± 0.08 | -0.33 ± 0.008 |
| 2010 | 13542.04 ± 562.26 | 5816 | 2.33 | -431.79 ± 9.74 | -3.09 ± 0.07 | -0.31 ± 0.007 |
| 2015 | 13082.14 ± 555.72 | 5875 | 2.23 | -459.90 ± 6.54 | -3.40 ± 0.05 | -0.68 ± 0.010 |
| Total | – | – | – | -1369.11 ± 27.68 | -9.47 ± 0.19 | -0.38 ± 0.013 |

The glacier covered area, number and average size on the north slope are smaller
than that of the south slope as seen in Table 6. The glacier covered area retreated by
413.60 km$^2$, which corresponds to an annual percentage of about 0.47% a$^{-1}$ from 1990




to 2000. Glacier area loss and rate in the second period (2000–2010) were
significantly higher than for 1990–2000. In the third period (2010–2015), the glacial
area reduction is about 302.07 km$^2$, and the annual percentage of area change
(0.77% a$^{-1}$) is greater than the first two periods.

**Table 6.** Glacier area distribution and changes of northern in the Himalaya from 1990 to 2015

| Year | Area (km$^2$) | Number | Average size (km$^2$) | Variation (km$^2$) | Variation rate (%) | APAC (% a$^{-1}$) |
|---|---|---|---|---|---|---|
| 1990 | 8778.02 ± 413.88 | 6561 | 1.34 | – | – | – |
| 2000 | 8364.42 ± 409.83 | 6674 | 1.25 | -413.60 ± 4.05 | -4.71 ± 0.05 | -0.47 ± 0.005 |
| 2010 | 7896.10 ± 397.35 | 6837 | 1.15 | -468.32 ± 12.48 | -5.60 ± 0.15 | -0.56 ± 0.015 |
| 2015 | 7594.03 ± 388.56 | 6883 | 1.10 | -302.07 ± 8.79 | -3.83 ± 0.11 | -0.77 ± 0.022 |
| Total | – | – | – | -1183.99 ± 25.32 | -13.49 ± 0.29 | -0.54 ± 0.019 |

Compared with the distribution and variation characteristics of glaciers on the
south and north slopes calculated in different periods (Table 5 and Table 6), which
were quite different. The glaciers are mainly distributed on south slope, which was
14451.25 km$^2$ accounting for 62.21% of the total area in 1990. Although the glacier
covered area on south slope is large, the number is small, with larger sizes. Previous
studies have shown that the covered area and number of glacier are influenced by
mountain toward, water vapor conditions and topography, and the positive difference
of glaciation determined the sizes of the glacier (e.g. Su et al., 1993; Yin, 2012). The
Himalaya represents an E-W striking, the south and north slopes are relatively wide,
which is conducive to glaciers development. The south slope is affected by the
monsoon and the large amounts of moisture bring out abundant precipitation, and the
positive difference of glaciation is lager, resulting in a southerly orientation
distribution, and the average size is larger, which showed that the abundant
precipitation brought by the southwest monsoon. However, the south slope is steep,
and the ridges and peaks are developed resulting in the relatively few in number.

The north slope is connected to the Tibetan Plateau and the mountains are
relatively flat where the glaciers are small, and the ridges and peaks aren't developed.
The numbers of glacier show a northward advantage and the average size of the
glaciers is small. In addition, the high mountains of the Himalaya hinder warm and
humid air mass from southwest direction to the north, resulting in less precipitation on
the north slope, which is not good for the development of glacier. Therefore, the
northern slope has a small distribution of glaciers. Previous studies for the south and
north slope of the parts in the Himalaya showed that the glaciers are generally
retreating. Kulkarni et al. (2007) used a number of Indian Remote Sensing satellites to
estimate glacial retreat for 466 glaciers in Chenab, Parbati and Baspathe basins in the
south slope of the Himalaya and found that the annual shrinkage rates are 0.56% a$^{-1}$,
0.48% a$^{-1}$ and 0.53% a$^{-1}$ for 1962–2001, respectively. Bolch et al. (2011) based on
multitemporal space imagery to investigate the glacier changes in the Khumbu Himal,
Nepal and the result showed that an average area loss of ice coverage by 5% from
1962 to 2005, with the highest retreat rates occurring between 1992 and 2001.
Bhambri et al. (2011) mapped glacier outlines for the Garhwal Himalaya in the south





slope using Corona and ASTER satellite images and found glacier area loss
0.15±0.07% $a^{-1}$ during the period of 1968–2006. Yin (2012) depended on the first
China and Nepal Glacier Inventory as well as remote sensing data to analyze glacier
variation characteristics on the south and north slopes in the Mt. Qomolangma and
found the average annual shrinkage rate of glaciers on the north slope was 0.25% $a^{-1}$,
which is higher than the south slope (0.23% $a^{-1}$) and it is consistent with our research.
*4.1.3. Glacier distribution and changes in the western, middle and east parts*
The glaciers were mainly distributed in the western Himalaya (Table 7), and is
about 11,551.69 km$^2$, accounting for 49.73% of the total glacier area in 1990. While
ice coverage is only 3092.83 km$^2$, representing 13.31% of the total glacier area in the
eastern part in 1990.
**Table** 7. ice coverage in the different regions of the Himalaya for 1990–2015 (unit: km$^2$)

| Year | 1990 | 2000 | 2010 | 2015 |
| --- | --- | --- | --- | --- |
| Western | 11551.69 ± 546.82 | 11117.70 ± 541.50 | 10671.78 ± 529.02 | 10242.10± 518.95 |
| Middle | 8584.75± 332.91 | 8267.95 ± 324.35 | 7953.78 ± 316.43 | 7711.22 ± 313.46 |
| Eastern | 3092.83 ± 117.55 | 2952.60 ± 115.98 | 2812.58 ± 114.16 | 2722.85 ± 111.87 |
| Total | 23229.27 ± 997.28 | 22338.25 ± 981.83 | 21438.14 ± 959.61 | 20676.17 ± 944.28 |

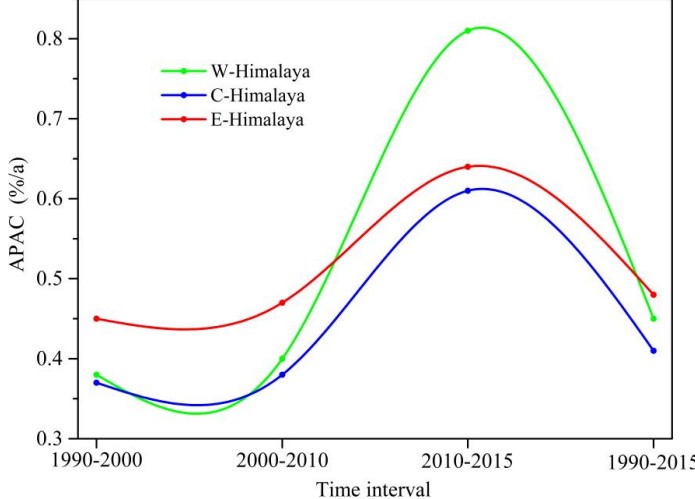

**Fig. 8.** The APAC in different regions of Himalaya from 1990 to 2015
The APAC was about 0.48% $a^{-1}$, 0.41% $a^{-1}$ and 0.45% $a^{-1}$ during the period of
1990–2015 in the eastern, middle and western parts, respectively. Glaciers shrank in
different regions, which showed that the glaciers have accelerated retreat in different
periods (Fig. 8), especially for the western Himalaya.
The annual average retreat rate is more rapidly in the eastern Himalaya than those
in the western and central parts in 1900–2015. Bolch et al. (2019) found that the
glacier area change in the eastern Himalaya glaciers have tended to shrink faster than
glaciers in the central or western Himalaya. In addition, Yao et al. (2012) analyzed the
variations of glacial area and mass balance in the eastern, central and western parts of
the Himalaya from 2005 to 2010 and found that the area of glaciers in the Himalaya
showed a trend of shrinking during the study period, and the annual average retreat
rate in the eastern part was the largest, followed by the western section, and the
central part is the smallest and the mass balance in different regions also showed
similar characteristics, which is consistent with our research.
*4.2. Glacier distribution and retreat in different elevation zones*
Analysis of the glacier hypsography showed that the majority of glaciers are
distributed at altitudes from 4,800 m to 6,200 m with an area percent of approximately
84% in 1990 and the highest ice coverage ranged is 5,200 m and 5,600 m (Fig. 9).
The ice coverage gradually decreases with altitude above 5,600 m. While the altitudes
exceed 7,000 m, the ice coverage only accounts for about 1.5% of the total area. The
possible reason is that within this height range, the mountain has a small distribution
area, and the cutting intensity is large, the terrain is broken, and the steep terrain is not
conducive to glacial development. During the extraction of the glacial boundary, we
also found some snow free bedrocks on high altitude areas.
The total area of the mountains above 4,000 m of the Himalaya is about $1.59 \times 10^{5}$
km$^{2}$, which provides a good topographical condition for glacial development. The
distribution of the Himalaya with altitude is consistent with the characteristics of the
glaciers. That is, there was a normal distribution between mountain area and elevation,
reaching the maximum at an altitude of 4,800~5,200 m (Fig. 9).

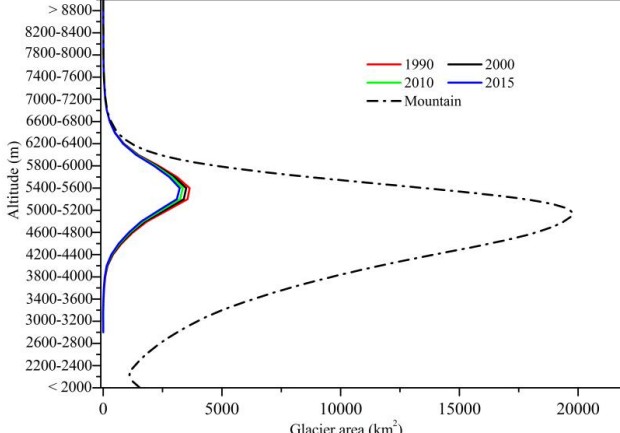


**Fig. 9.** Ice coverage and mountain distribution at different elevations between 1990 and 2015

Glacial development is affected by topographical terrain and climatic conditions
(Li et al., 1986). The Himalaya provides favorable terrain for the development of
glaciers. In addition, the impact of the climate should not be underestimated. Shi et al.
(1982) considered that the climate gradually developed toward the wet and cold with
the elevation within certain range, which is favorable for the development of glaciers.
However, the precipitation showed a decreasing trend as altitude rises further and the
climate is gradually developing towards dry and cold, which inhibits the development
of glaciers to some extent.



There was a normal distribution between glacier area and elevation, and ice coverage reached the maximum at altitudes of 5,200~5,600 m. It can be seen that, the temperature and gradually decreases, and the precipitation gradually increases within a certain range with the elevation, which is benefit to glaciers developed. Combined with the vertical distribution of the Himalaya, although mountains reach the maximum at 4,800~5,200 m, the glaciers only have about 19% in this interval, and ice coverage reaches the maximum at 5,200~5,600 m. The possible reason is that 4,800~5,200 m is not the upper limit of the wet and cold, and 5,200~5,600 m may be the turning point about dry and wet, which is the "second major precipitation zone" in the Himalaya. Depended on the supply of the "second largest precipitation zone", favorable topography and low temperature conditions, the glaciers are developed in this area. Li et al. (1986) though the latent heat generated by condensation is a heat source for the strong updraft and is also the main reason for the formation of high altitude and topographical rain caused by local circulation, which is an important supply for many mountain glaciers on the Tibetan Plateau. Xie and Su (1975) believed that this local circulation has formed a distinct "second largest precipitation zone" on the southern margin of the Tibetan Plateau. The most typical case is the Everest region of the Himalaya. In addition, Yasunari and Inoue (1978) observed the existence of "second largest precipitation zone" in the Himalaya, which is also the result of the local circulation of high mountains in summer, and pointed out that "second major precipitation zone" is above 5,000 m.

The glacier areas have not change significantly above 6,600 m in the past 25 years. Therefore, we only counted the range of 3,000 m to 6,600 m (Fig. 10). The area of glaciers in all altitudes has decreased and reached the maximum at 5,200~5,400 m, which may be related to the development of glaciers between 5,200 and 5,600 m. Analyzed of glacial retreat rates at different altitudes showed that it occurred below 4,600 m. There were two characteristics about the trend of glacial area change with altitude: (1) the change is more complicated from 3,000 m to 3,800 m, which was increases first, then decreases and then increases and they reach 41% and 37% in 3,000~3,200 m and 3,200~3,400 m. Although the absolute changes were small in the above two height ranges, the overall distribution area of the glacier was also small, resulting in the glacier retreat rate larger. Further up, at an altitude of 3,400~3,600 m, the glacier retreat rate has dropped significantly. It is found that the glacier retreat in this range has little difference with the range of 3,000~3,400 m, but the glacier distribution area is the three times of the glacier at 3,000~3,400 m, resulting in a significant decrease in the rate of glacial retreat in this region, (2) ice coverage retreat rate fluctuates decline with the elevation in 3,800~6,600 m.

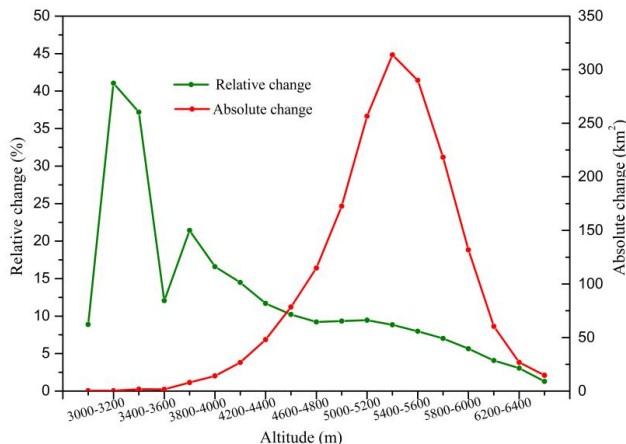

**Fig. 10.** glacier area variations at different elevations during the period of 1990–2015

*4.3. Glacier distribution and variations in different forms*

*4.3.1. Glacier distribution and retreat of different morphological types*

Glaciers of the Himalaya belong to mountain glaciers. According to the types of mountain glaciers and combing with the three-dimensional image display features of Google Earth, we divided the glaciers of the study area into hanging glacier, valley glacier, cirque glacier, cirque-valley glacier and ice cap. The distribution of the number and area of different types of glaciers in the Himalaya was studied in 1990, and we also analyzed the glacier variations of different morphological types between 1990 and 2015.

To verify the extraction accuracy of the various morphological types glaciers, we compared the results extracted by Google Earth in the Namurani and the Narangalkang regions of the Himalaya with the results of field measurement by Li (Table 8), and the results showed that the extraction by Google Earth in this study are highly consistent with the field measurement, which can meet the needs our research.

**Table 8.** Glacier number in the Naimona'nyi and the Narangalkang regions of the Himalaya

| morphological types | the Naimona'nyi | | the Narangalkang | |
|---|---|---|---|---|
| | Li et al (1986) | Our research | Li et al (1986) | Our research |
| valley glacier | 5 | 5 | 1 | 1 |
| hanging glacier | 37 | 32 | 43 | 35 |
| cirque glacier | 5 | 4 | 14 | 14 |
| cirque-valley glacier | 11 | 12 | 4 | 4 |
| ice cap | 0 | 0 | 0 | 0 |
| Total | 58 | 53 | 62 | 54 |

As shown in Table 9, the largest number is hanging glacier, and there are 7883, contributing 64.56 % of the total number in 1990, whereas the number of ice cap is the fewest and represents 0.16%. The largest ice coverage is valley glacier, which accounts for ~53.33% of the total glacier area in 1990. The valley glaciers are on average about 9.82 km$^2$ in size and hanging glacier is the smallest, which is only





about 0.61 km$^2$. Although the number of the valley glacier rank third in the Himalaya, ice coverage is very large, nearly double the total area of other types. Valley glacier is the most important type in the Himalaya and it has the following features: the firn basin is relatively wide and the rear wall is steep, and the aretes and peaks are developed; there are glacial rapids below the accumulation area; there are surface rivers and subglacial rivers in the ablation area (Fig. 11) and the moraine is relatively developed in glacier tongue.

**Table 9.** Glacier area and number in different morphological patterns of the Himalaya in 1990

| morphological types | 1990 | | | shrinkage rate for 1990–2015 (%) |
|---|---|---|---|---|
| | Number | Area (km$^2$) | Size (km$^2$) | |
| valley glacier | 1261 | 12387.63 | 9.82 | 6.50 |
| hanging glacier | 7883 | 4782.33 | 0.61 | 20.04 |
| cirque glacier | 1156 | 1093.01 | 0.95 | 18.03 |
| cirque-valley glacier | 1891 | 4946.73 | 2.62 | 12.19 |
| ice cap | 20 | 19.57 | 0.98 | 14.11 |

Glacier size strongly affects the loss percentage in glacier area and there was a negative correlation between the shrinkage rate and the average size of glaciers between 1990 and 2015. The larger glaciers have the smaller the retreat rate. The average size of the valley glaciers is the largest, and the glaciers of this type have the smallest retreat rate, only 6.50% in the past 25 years. In comparison, the size of the ice cap is the smallest, but the glacial area retreat rate is the largest, which is about 20.04%, followed by the cirque glacier, ice cap and cirque-valley glacier, equals 18.03%, 14.11% and 12.19%, respectively.

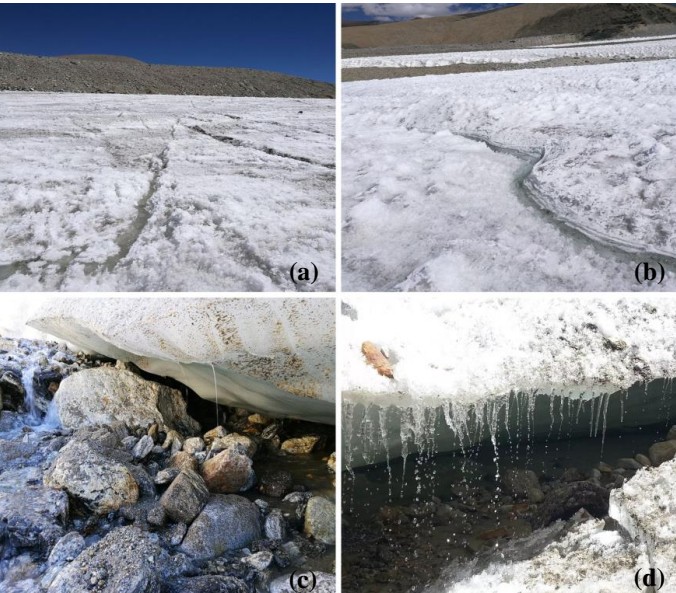

**Fig. 11.** (a) and (b) the surface rivers of Zhongni Glacier in 2016; (c) the Subglacial river of 5Z342B0021 Glacier and (d) Subglacial river of Zhongni Glacier in 2016


### 4.3.2. Glacier distribution and changes of debris-covered and debris-free ice

The debris-covered glaciers are developed in the Himalaya, especially for the
southern slope. Previous study showed that debris-covered glaciers are about 25% of
the total glacierized area in the d Himalayan ranges (Bajracharya and Shrestha, 2011).
Su et al. (1985) though there are several factors to form debris-covered glaciers: (1)
avalanche. Due to the avalanche, a large amount of debris is carried on the hillside
forming abundant inner moraine. However, the shallower inner moraine will be
exposed to the surface because of the melting of the ice surface with glacier
movement and this kind of surface moraine is mostly sub-angular; (2) glacier
movement. During the downward movement of the glaciers, some of inner and
bottom moraines move to the surface of the glacier to form the moraine, which has
better roundness and smaller size; (3) Cold weathering. Some rock masses on the
slopes on both sides of the glacier collapsed to the surface of the glacier due to the
cold weathering, forming surface scorpions, which are mostly angular blocks; (4)
glacial convergence. After the glaciers meet, the lateral moraine of glaciers becomes
middle moraine, which makes the surface of the moraine distributed in strips. Scherler
et al. (2011) though rocky debris are linked with hillslope-erosion rates, which are
related to hillslope angle and therefore the formation of debris-covered glaciers are
linked to steep (>25°) accumulation areas. Accumulation areas in the Karakoram are
relatively steep (meanhillslope angles 25°–35°), and debris are frequent. Many glaciers
have heavily debris in Himalaya-Karakoram, a further consequence of the steep rocky
terrain and avalanche activity (Bolch et al., 2012). Most glaciers in the Himalaya,
Nyainqêntanglha and Hengduan Mountains have heavily debris, and these areas are
mainly affected by the monsoon where the precipitation is abundant, which makes the
mountains more humid, thus strengthening the weathering of the mountain rocks and
the weathered rocks is gradually transported to the glacier surfaces form debris under
the influence of avalanches, which showed that the debris are the combination result
of topography, glacial size, climate and avalanche.
When the surface debris thickness is greater than 0.02 m and the internal debris is
quite developed, it not only hinders the heat transfer, but also has an important
influence on the hydrostatic pressure, ice density, ice temperature and ice stress field
in the middle and lower parts of the glacier. The heat insulation effect is very obvious,
which has a strong inhibitory effect on the ice surface ablation (e.g. Mattson et al.,
1993; Lu et al., 2014). Su et al. (1985) showed that when the thickness of the debris
exceeds 0.1 m, the amount of glacial ablation can be effectively reduced by about
10%. Conversely, when the debris is thin (≤0.02 m), it can absorb more solar
radiation, and the presence of the surface debris can accelerate the melting of the
glacier. In summary, the rate of ablation of debris-covered glaciers is not necessarily
lower than that of debris-free glaciers. The main reasons include: (1) the rate of
glacial ablation covered by debris may also be affected by the ice front lake and ice
cliffs. Glacial meltwater formation of ice lakes and ice cliffs can transfer heat to the
glacier tongues, thus accelerating ablation (e.g. Bolch et al., 2012; King et al., 2017);
(2) the surface flow rate of debris-covered glaciers is lower than that of debris-free





glaciers. Therefore, the ablation of the ice surface can only be replenished by a small
amount of ice from the upstream for debris-covered glaciers, resulting in ablation rate
of this type of glaciers higher than that of debris-free glaciers (Gardelle et al., 2012).
The debris of the Himalaya is relatively developed. Can the presence of debris in
this region inhibit glaciers melting? What are the distribution characteristics of debris?
What is the upper elevation of debris? Are they mainly distributed on gentle hillside
or steep areas? In order to solve these problems, we divided the glaciers of the
Himalaya into debris-covered glaciers and debris-free glaciers, and studied the
distribution and variation characteristics of two type glaciers in 1990-2015 (Table 10).
**Table 10.** Glacier area and number in different situations of the Himalaya for 1990−2015

| Year | Debris-covered glacier | | | Debris-free glacier | | |
|---|---|---|---|---|---|---|
| | Area (km$^2$) | Number | Average size (km$^2$) | Area (km$^2$) | Number | Average size (km$^2$) |
| 1990 | 10269.37 | 749 | 13.71 | 12959.90 | 11462 | 1.13 |
| 2015 | 9733.22 | 754 | 12.91 | 10942.95 | 12004 | 0.91 |

The number of debris-covered glaciers in the Himalaya is relatively small, which
is only 749 and 754 in 1990 and 2015, respectively, accounting for 6.13% and 5.91%
of the total number. Although the number of debris-covered glaciers is small, its
distribution area is relatively large, accounting for 44.21% and 47.07% of the total
glaciers in the corresponding year. By comparing the average size of the two types of
glaciers, we found the debris-covered glaciers are about 13.71 km$^2$ and 12.91 km$^2$ in
1990 and 2015, respectively. Compared to the debris-covered glaciers, the distribution
area and number of the debris-free is large, resulting in a smaller average size, which
is only 1.13 km$^2$ and 0.91 km$^2$ in 1990 and 2015, respectively.
To investigate whether the debris of the Himalaya can inhibit the glacier melting,
we analyzed the area shrinking rate of two types of glaciers. The result showed that
the total area loss of the debris-covered glaciers is about 536.15 km$^2$, with a rate of
area retreat 5.22% for 1990–2015 and the debris-free glaciers are 2016.95 km$^2$ and
15.56%, respectively. The ice area loss of the debris-free glaciers is three times that of
the debris-covered glaciers, which shows that the debris of the Himalaya can inhibit
the glacier melting to a certain extent. Immerzeel et al. (2014) found that when the
debris thickness in the Himalaya is greater than 0.4 m, the ablation rate at the end of
the glacier is significantly reduced. In addition, the average size of the debris-covered
glaciers in the study area is 12 times that of the debris-free glaciers, which may also
be an important factor for the small ice area loss of the debris-covered glaciers.
The altitude and slope of debris of the Himalaya in 1990 were showed in Fig. 12.
There was a normal distribution between debris area and elevation, and the lower
limit of the distribution is about 3,000 m and mainly concentrated on the range of
4,400~5,200 m, with approximately 61.14%. The debris with 4,600~4,800 m exhibit
the largest area, accounting for 16.67% of the total area in 1990 and the debris
coverage is less in the lower than 3,800 m and higher than 6,000 m, only contributing
to 2 % to the total area. The mean slope of debris in this region ranges from 0 ° to 60 °
and mainly distributes 0-20 °, contributing to 68.81% of the total area in 1990 (Fig.





12b). Debris with mean slopes of 5-10 °(covering an area of 26.13% in 1990) exhibits
the largest area and the slope is larger than 40 °, and the distribution of debris in the
Himalaya is small, accounting for only 6.18%. With the increase of the slope, the
coverage area of the debris gradually decreases. When the slope is larger than 60 °, the
area is only 0.43%. In summary, the debris of the study area is mainly distributed in
the 4,400~5,200 m and gentle zones.

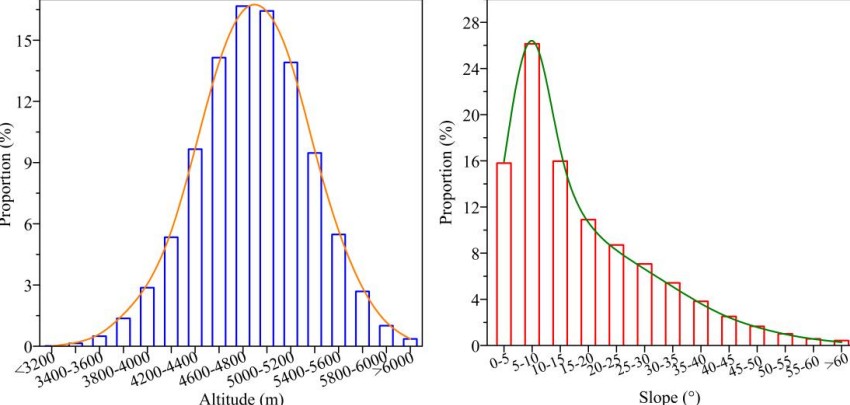


**Fig. 12.** Moraine distribution in 1990. (a) the altitude and (b) the slope
*4.3.3. Glacier distribution and variations of temperature glaciers and continental*
*glaciers*
Most glaciers in the eastern and southern slope of the Himalaya belong to the
"summer-accumulation type" or temperature glaciers, gaining mass mainly from
summer-monsoon snowfall (Bolch et al., 2012), and continental glaciers are widely
distributed from northern slope of the western Himalaya to the central Himalaya. Due
to differences in hydrothermal conditions, physical properties and ice formation
between temperature glaciers and continental glaciers, the response processes and
mechanisms for climate change are also different. With global warming, the
distribution and variations of these two types of glaciers in the Himalaya and their
research on the response to climate change are of great significance (Table 11).
**Table 11.** The moraine and continental glaciers of the Himalaya from 1990 to 2015

| Type | Year | Area (km$^2$) | Number | Average size (km$^2$) | Variation rate (%) | APAC (%/a) |
|---|---|---|---|---|---|---|
| Temperature | 1990 | 16340.18 | 7007 | 2.33 | – | – |
| glaciers | 2015 | 14732.94 | 7284 | 2.02 | -9.84 | -0.39 |
| Continental | 1990 | 6889.09 | 5204 | 1.32 | – | – |
| glaciers | 2015 | 5943.23 | 5474 | 1.09 | -13.73 | -0.55 |

There are 7007 glaciers of the temperature glaciers with a total area of 16340.18
km$^2$, contributing to 57.38% and 70.34% of the total number and area in 1990. It can
be showed that the abundant monsoon precipitation in the Himalaya provides good
conditions for glacial development.
Based on our data, the glaciers retreat both types of glaciers from 1990 to 2015,



but the annual percentage of area change was not consistent. The temperature glaciers
have decreased 1607.24 km$^2$, and the APAC was about 0.39% a$^{-1}$. Compared with the
temperature glaciers, the area loss of the continental glaciers is less, which is only half
of the temperature glaciers, but the shrinkage rate (0.55% a$^{-1}$) was larger than the
former. In addition to the glacier area loss in the Himalaya, the average size of
temperature and continental glaciers in this study area is also decreased. The average
size of the temperature glaciers decreased form 2.33 km$^2$ in 1990 to 2.02 km$^2$ in 2015,
while the continental glaciers change from 1.32 km$^2$ to 1.09 km$^2$ for 1990–2015. The
main reason for the reduction of the average size is probably the area loss and the
fragmentation of glaciers in the Himalaya.
The APAC of temperature glaciers is smaller than that of the continental glaciers,
which is contrary to the results of previous studies. Su et al. (2015) analyzed the
typical glaciers of the Tianshan Mountains and the Alps and compared the changes in
mass balance between continental glaciers and temperature glaciers, and the results
showed that the inter-annual variability and loss in mass balance of the temperature
glaciers is significantly higher than that of the continental glaciers and the temperature
glaciers are more sensitive to climate change. Li (2015) studied the variations of the
temperature glaciers in western China and recorded the retreat rate of the temperature
glaciers is relatively large and the retreat is more severe due to the lower elevation of
the mountain and the smaller size of the glaciers in the Gangri and Yulong Snow
Mountain of the Gongga Mountains. Wang (2017) compared the temperature glaciers
and continental glaciers in Tanggula Mountain and found that the temperature glaciers
are more sensitive to climate change because of their smaller size and lower elevation.
It can be seen that the types, sizes and elevations of glaciers played an important role
to glacier shrinkage. The ice coverage loss of the temperature glaciers in the Himalaya
is larger. The reason may be related to the sizes of glaciers. The study shows that the
average size of temperature glaciers is significantly larger than that of continental
glaciers, and the former is about 2.33 km$^2$ and the latter 1.32 km$^2$. On the other hand,
it may be related to the coverage of debris. The pervious study showed that the debris
in the Himalaya can inhibit the glaciers melting to some extent and the debris of the
southern slope of the Himalaya was relatively developed. To study whether the debris
have an effect on the temperature glaciers and continental glaciers, we removed the
debris of the Himalaya and explored the area loss of the temperature and continental
glaciers in the Himalaya without debris coverage and the result shown in Table 12.
**Table 12.** Glacier distribution and changes in moraine and continental glaciers about debris-free
ice of the Himalaya between 1990 and 2015

| Type | Year | Area (km$^2$) | Area loss (km$^2$) | Variation rate (%) | APAC/ (%/a) |
|---|---|---|---|---|---|
| Temperature | 1990 | 14064.50 | – | – | – |
| glaciers | 2015 | 12015.70 | -2048.80 | -14.57 | -0.58 |
| Continental | 1990 | 6544.02 | – | – | – |
| glaciers | 2015 | 5604.15 | -939.87 | -14.36 | -0.57 |


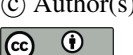



Ice area loss rate and APAC of the temperature glaciers are larger than those the
continental glaciers regardless of debris. The APAC of the temperature glaciers is only
larger than 0.01% for the continental glaciers. To eliminate the errors caused by the
visual interpretation, we carefully examined the results of the glacier boundaries. In
summary, the debris and the glaciers average sizes in the Himalaya may have an
important impact on the annual shrinkage rate. The temperature glaciers in the
Himalaya are more sensitive to climate change without debris.
Solar radiation, topography, temperature, precipitation, debris, glacial sizes, and
surface morphology are important factors to influence glacier area loss (e.g. Scherler
et al., 2011; Yao et al., 2012). Although the above factors have an impact influence to
glacier changes, these factors have different spatial and temporal scales. For example,
the temperature can affect the glacier area change on a larger space-time scale. In
contrast, other factors can affect glacier variations on a small time and spatial scale
(Shi et al., 2000). Among all the affecting factors, climatic factors area probably the
most important. Temperature and precipitation have a close relationship with glacier
changes (e.g. Gao et al., 2000; Liu et al., 2006; Xu et al., 2008). Zhao et al. (2004)
examined change of climate 50 meteorological stations in the Tibetan Plateau for
1976–1997 and the results showed that the Tibetan Plateau has shown a trend of
warming in the past 30 years, and the warming trend was greater in the cold season
(August to March) than in the warm season (April to September). Ren et al. (2004)
compared the Dingri and Nyalam weather stations in the central of the Himalaya and
found that the temperature rises in the dry areas is significant than that in the wet
regions. The reason for the temperature glaciers area more sensitive to climate change
probably related to temperature, precipitation and glacier's own factors. (1)
Temperature. The temperature glaciers in the Himalaya are mainly affected by the
southwest monsoon in summer, so the ice-temperature is higher and the warming rate
is opposite. (2) Precipitation. Previous studies have shown that the Indian monsoon
has weakened since the 1950s, while the westerly has shown an enhanced trend
(Gardelle et al., 2013). The possible consequence is that the winter precipitation
increases in westerly region and the summer precipitation reduces in monsoon region.
(3) The characteristics of glacier's own factors. The temperature glaciers are mostly
located in abundance precipitation region where the elevation of glacier tongue is
often low (Fujita and Nuimura, 2011), and the ice temperature is also higher than that
of the continental glaciers, and the temperature glaciers belong to the
"summer-accumulation type", gaining mass mainly from summer-monsoon snowfall,
while continental glaciers belong to the "winter accumulation type", that is, summer
melts and winter accumulates. With the temperature rises, the proportion of rainfall in
precipitation increases, and the solid precipitation falling on the surface of the glacier
decreases and extends the melting period. Without a snow cover in summer, surface
albedo is much lower and melt is further increased (Bolch et al., 2012). In recent years,
Scholars have investigated the temperature glaciers of the Hengduan Mountain and
found that the ice structure of the glacier in this region has significant changes. Ice
crevasse and ice holes area widely distributed and the number increases, and ice falls



are frequently collapsed. The degree of ice fragmentation is more serious, which have
seriously damaged the glaciers integrity and adaptive mechanism, and increased the
glaciers melting area resulting in the intensification of the glacier melting and
shrinking (e.g. Li et al., 2009; Liu et al., 2014). In addition, the altitudes of the
temperature glaciers tongues is lower, and the ice temperature is higher (Liu et al.,
2013), which is more sensitive to temperature rise. It can be seen that the temperature
glaciers are more sensitive to climate change likely to be the result of the combination
of temperature, precipitation and glacier characteristics.
**5. Conclusions**
We combined remote sensing data and ASTER GDEM to construct glacier
inventory for the entire Himalaya Range that do not have sufficient observational data
records, and to quantity glacier area and changes in different regions and elements.
Spatial trends of glacier area distribution and changes in the past 25 years include:
(1) Glacier area change amounts to 0.44% $a^{-1}$ during the period of 1990-2015,
with a higher retreat rate in the last 5 years (0.71% $a^{-1}$ from 2000 to 2015) compared
to the previous period (0.38% $a^{-1}$ and 0.40% $a^{-1}$ during the periods 1990-2000,
2000-2010, respectively), small and steep glaciers are more sensitive to climate
change and smaller glaciers have disappeared;
(2) Glaciers are mainly distributed in south slope with an area about 14451.26 $km^2$
in 1990, accounting for ~62.21% and the average annual shrinkage rate of the glaciers
on the north slope (0.54% $a^{-1}$) is greater than that on the south slope (0.38% $a^{-1}$) ;
(3) Larger area distribution in the western of the Himalaya and eastern is
minimum, the glaciers retreated in the western, middle and eastern of the Himalaya
during 1990-2015. The eastern was fast and the middle was slowest.
(4) The glaciers were mainly distributed at approximately 4,800~6,200 m a.s.l.
and the largest glaciers in the area showed the elevation of 5,200~5,600 m a.s.l. which
may be the turning point about dry and wet, which is the "second major precipitation
zones" in the Himalaya.
(5) Higher rates of retreat for debris-free glaciers (15.56%) on a glacier-by-glacier
basis, compared to debris-covered glaciers (5.22%) in the last decades;
(6) The largest ice coverage and average size is valley glacier, which is the most
important type in the Himalaya and has the following features: the firn basin is
relatively wide and the rear wall is steep, and the aretes and peaks are developed, and
avalanches occur frequently; there are glacial rapids below the accumulation area;
there are surface rivers and subglacial rivers in the ablation area; the moraine is
relatively developed in glacier tongue.
**Acknowledgments**
This research was funded by National Natural Science Foundation of China (Grant No. 41801063),
the Science and Technology Research Program of Chongqing Municipal Education Commission
(Grants No. KJQN201800541, KJQN201900548), Humanity and Social Research Project of
Education Ministry to Young Scholars (Grant No. 16YJCZH061), Tianyou Youth Talent Lift
Program of Lanzhou Jiaotong University, and China postdoctoral science foundation



(2019M660092XB). We thank the United States Geological Survey (USGS), National Earth System Science Data Center and the Geospatial Data Cloud for providing data, and the China Meteorological Data Sharing Service System for providing the meteorological data.

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
