# Peer review of "Glacier variations in the Himalaya from 1990 to 2015 based on remote sensing"

_The Cryosphere, 2019_

## Referee Comment (RC1) · Anonymous Referee #1 · 6 May 2020

Dear Authors

I think your submission is not ready for review, the English (grammar and wording) is simply too bad and the text is unreadable. Please forward the study to a native speaker for proof-reading and correction before re-submitting it. Please also note that the cited references and datasets are partly very old and outdated. For example, since several years the 30 m SRTM1 and GDEM2 are available and there is also the ALOS AW3D30 DEM. Moreover, the GAMDAM2 inventory by Sakai (2019)* is providing an excellent glacier outline dataset for entire High Mountain Asia, but the study is not even cited. From Fig. 5 and 7 I see that several smaller glaciers have not been mapped. Such random omissions need to be explained. As a minor point, please also improve the layout by inserting empty lines before section headings and before/after figures and

tables if a re-submission is considered.

*https://doi.org/10.5194/tc-13-2043-2019
* * *

---

## Referee Comment (RC2) · Anonymous Referee #2 · 26 May 2020

Based on the title, the paper by Ji et al aims at presenting glacier variations in the Himalaya in the last decades. However, it is not clear what exactly the authors mean by glacier variations, and why the need for such a study while there are already regional studies on glacier changes, and lots of glacier inventories to be used. In the introduciton the authors state that there is a need to study the glacier distribution and changes across the Himalaya, and they propose a new set of glacier extents to investigate changes from 1990 to 2005, along with glacier distributions.

First, to me, it seems that this is a duplicate effort to previous work done in the region by past inventories, including the revised GADNDAM as well as other regional inventories. The authors would first need to present outstanding issues with the existing inventories to justify yet another study. Second, as we know there is considerable effort needed

to produce high quality inventories - and the methods used in this paper do not seem superior over the existing ones.

The English language throughout the paper is insufficient, and needs lots of improving. It is hard to estimate the value of the research conducted because the paper is very poorly written and not structured. Unfortunately I do not consider this submittable; the objectives need be be clarified and the writing needs much improving in order to assess the value of this paper.

Some general comments by section:

Introduction: The introduction is rambling and not to the point. It is not necessary in a journal specialist in cryosphere to explain what cryosphere is, for ex., or to mix lots of concepts, eg mass balance and SLR estimates, then water resources and formation of the Himalaya.

Study area: again this is rambling. Glacier classification into "continental" and "temperature glaciers" is erroneous, and references are outdated. References about climate data are also outdated. This is not a climate paper so such details are meaningless. Showing pictures of Everest in figure 1 does not bring anything to the paper.

Data and Methods: No new methods were used here, and the authors state that for some years they averages +/- 2 years. It makes it hard to know then what year the glacier outlines represent when for some areas various years of imagery were used. Reporting cloud cover for each scene is irrelevant since cloud cover can be outside the glaciers. It becomes clear only later on that the authors derived ne set of outlines per decade. In this case, why not use RGI which is also from $\sim$2000 and is a compilation? The authors would need to show the superiority of their approach over existing outlines, but this is not done in the paper.

The SRTM vs ASTER comparison version is also inadequately written-

Glacier mapping also is based on established methods (band ratios) and this section

does not need almost one page to describe. Just a technical detail, band ratio of 1 seems very low for OLI, while 1.8 for TM seems acceptable; the authors do not comment on this. Also, the authors refer to the 2nd Chinese inventory but do not add a reference, nor how it was used. Debris covered glaciers, as it seems, were mapped manually but the description is fuzzy and there is nothing innovative here.

Error estimates are based only on a single glacier and seem incomplete. A study of such spatial extent would need a much more thorough error and uncertainty section.

Results -

The authors present glacier changes across the study area with respect to various factors: part of the range, elevation bin, type of glaciers etc. While this is of possible interest, especially with respect to the spatial distribution of glacier changes, the information presented is hard to distill and very dense. This needs much more synthesis.

Also, the changes are referred to most of the times as simply "changes"- it should be mentioned when the changes are in glacier area, length or height etc.

Some concepts need much more development, for example debris cover. For example l 492 - 495 the authors mention the melt inhibition due to thin debris- however in the recent years there have been a number of publications which point at the presence of supra-glacial lakes and ice cliffs and their effect on melting rates over thick debris. The references presented are also outdated. Also the authors claim they test the effect of debris on melting rates- "To investigate whether the debris of the Himalaya can inhibit the glacier melting" (l522) - but then they present area changes, while debris cover glaciers can get thinner yet display no area change. So area change as an indicator of surface melt is not an appropriate measure.

In general results are difficult to follow in the form they are presented. For example, the authors compare the glacier change analysis with other studies- but there is no reference as to which year, so this is quite meaningless without adding more details. For

example, they show only 1.3% difference with Guo et al (2015) - but it is unclear what "regional Himnalaya" means. Results on the glacier distribution across the Himalaya are mixed with glacier changes and this is all hard to follow, for example phrases such as " The total area of the mountains above 4,000 m of the Himalaya is about $1.59\times105$ km2 which provides a good topographical condition for glacial development" are not very meaningful, I am not sure what the authors mean. Same for l 373 to 381- is this relevant to the scope of the paper?

The authors go onto mention conditions for glacier formation- but this has nothing to do with the scope of the paper.

The rest of the results are hard to follow and the text is poorly written, so a proper review of the results and discussion cannot be provided in the current version of this paper.

---

## Author Comment (AC1) · 1 Jun 2020

Dear Anonymous Referee,

Thanks you very much for your comments suggesting our manuscript entitled "Glacier variations in the Himalayas from 1990 to 2015 based on remote sensing". We have corrected grammar and wording by a native speaker, and improved the layout by inserting empty lines before section headings and before/after figures and tables in the revised manuscript. Several smaller glaciers have not been mapped in Fig. 5 and 7, and we explained it in the revised manuscript lines 194-202 (In this process, we extracted glaciers with an area greater than 0.05 km2 (the remaining small scales areas are likely to be snow). In addition, due to the impact of the quality of remote sensing images in different periods (the quality of a certain period or several periods was not ideal), there were glaciers with areas greater than 0.05 km2 that we also did not count. Considering the interpretation accuracy, we found that the distribution area of such glaciers was small, and the impact on the entire Himalayas was low). In addition, we can't find the article about the GAMDAM2 inventory by Sakai (2019), can you send us the titl eof article or published journal? Many thanks again.

Best wishes

Qin Ji

Please also note the supplement to this comment:
https://www.the-cryosphere-discuss.net/tc-2019-297/tc-2019-297-AC1-supplement.pdf

---

## Author Comment (AC2) · 8 Jun 2020

Dear Anonymous Referee,

Many thanks for your comments. We also clarified the objectives in the revised manuscript and corrected grammar and wording by a native speaker. In addition, we edited the introduction and study area according to your suggestion.

Best wishes

Qin Ji

---

## Author Comment (AC3) · 26 Jun 2020

Dear Anonymous Referee,

Many thanks for your comments. We also clarified the objectives in the revised manuscript and corrected grammar and wording by a native speaker. In addition, we edited the introduction and study area according to your suggestion.

Study area: again this is rambling. Glacier classification into "continental" and "temperature glaciers" is erroneous, and references are outdated. References about climate data are also outdated. This is not a climate paper so such details are meaningless. Showing pictures of Everest in figure 1 does not bring anything to the paper.

Response: According to the advice of anonymous referee, we deleted the sentence "Glacier classification into "continental" and "temperature glaciers"". In order to indicate the distribution of glaciers in the southern slope and northern slope, we made details in climate. Showing pictures of Everest in figure is the purpose to intuitively demonstrate the highest peaks in the world.

Data and Methods: No new methods were used here, and the authors state that for some years they averages +/- 2 years. It makes it hard to know then what year the glacier outlines represent when for some areas various years of imagery were used. Reporting cloud cover for each scene is irrelevant since cloud cover can be outside the glaciers. It becomes clear only later on that the authors derived ne set of outlines per decade. In this case, why not use RGI which is also from 2000 and is a compilation? The authors would need to show the superiority of their approach over existing outlines, but this is not done in the paper.

Response: The purposes of this paper analyzed the glacier distribution and variation characteristics in the entire Himalayas, so the method of glacier boundaries extraction is not new. The RGI are based on the ASTER data and this paper we extracted glacier outlines used Landsat TM/ETM+/OLI in 1990, 2005 and 2015. In order to avoid the impact of data inconsistency, we obtained the glacier boundaries used Landsat data and we also corrected potential error by visual interpretation in our paper.

Glacier mapping also is based on established methods (band ratios) and this section does not need almost one page to describe. Just a technical detail, band ratio of 1 seems very low for OLI, while 1.8 for TM seems acceptable; the authors do not comment on this. Also, the authors refer to the 2nd Chinese inventory but do not add a reference, nor how it was used. Debris covered glaciers, as it seems, were mapped manually but the description is fuzzy and there is nothing innovative here.

Response: Thanks for the Referee's suggestion we have added some contents in the details of image thresholding for Landsat OLI imagery and marked them in red in lines 183-190. In order to make the Landsat OLI data match with the Landsat TM/ETM+ imagery, we set Landsat TM/ETM+ as a reference to make geometric correction to Landsat OLI scenes. We also add the reference for 2nd Chinese inventory in line 204 in the revised manuscript.

Error estimates are based only on a single glacier and seem incomplete. A study of such spatial extent would need a much more thorough error and uncertainty section.

Response: Many thanks for your advice and we have used the buffer method to calculate the accuracy in lines 245-247 in the manuscript.

Results –

The authors present glacier changes across the study area with respect to various factors: part of the range, elevation bin, type of glaciers etc. While this is of possible interest, especially with respect to the spatial distribution of glacier changes, the information presented is hard to distill and very dense. This needs much more synthesis. Also, the changes are referred to most of the times as simply "changes"- it should be mentioned when the changes are in glacier area, length or height etc.

Response: The "changes" in this paper is referred to the area change.

Some concepts need much more development, for example debris cover. For example l 492 - 495 the authors mention the melt inhibition due to thin debris- however in the recent years there have been a number of publications which point at the presence of supra-glacial lakes and ice cliffs and their effect on melting rates over thick debris. The references presented are also outdated. Also the authors claim they test the effect of debris on melting rates- "To investigate whether the debris of the Himalaya can inhibit the glacier melting" (l522) - but then they present area changes, while debris cover glaciers can get thinner yet display no area change. So area change as an indicator of surface melt is not an appropriate measure.

Response: The Referee's comments are good. However, the large area covered by glaciers in the Himalayas, it is more difficult to calculate the change in height, so we used area change as an indirectly indicator.

In general results are difficult to follow in the form they are presented. For example, the authors compare the glacier change analysis with other studies- but there is no reference as to which year, so this is quite meaningless without adding more details. For example, they show only 1.3% difference with Guo et al (2015) - but it is unclear what" regional Himnalaya" means. Results on the glacier distribution across the Himalaya are mixed with glacier changes and this is all hard to follow, for example phrases such as " The total area of the mountains above 4,000 m of the Himalaya is about $1.59 \times 10^5$ km$^2$ which provides a good topographical condition for glacial development" are not very meaningful, I am not sure what the authors mean. Same for l 373 to 381- is this relevant to the scope of the paper?

Response: Many thanks for the Referee's advice. We wanted to analysis the relationship between glaciers and topography, so we statistics the area in different elevation zones of the Himalayas mountain.

We appreciate for the Referee's warm work earnestly, and hope that the corrections will meet with approval.

Best wishes

Qin Ji

---

## Author Comment (AC5) · 2 Jul 2020

Q1: Please forward the study to a native speaker for proof-reading and correction before re-submitting it. A1: Thanks you very much for your comments suggesting our manuscript entitled "Glacier variations in the Himalayas from 1990 to 2015 based on remote sensing". We have corrected grammar and wording by a native speaker.

Q2: Please also note that the cited references and datasets are partly very old and outdated. For example, since several years the 30 m SRTM1 and GDEM2 are available and there is also the ALOS AW3D30 DEM. Moreover, the GAMDAM2 inventory by Sakai (2019)* is providing an excellent glacier outline dataset for entire High Mountain Asia, but the study is not even cited. A2: Many thanks for your advice, but we can't find

the article about the GAMDAM2 inventory by Sakai (2019), can you send us the title of article or published journal? Many thanks again.

Q3: From Fig. 5 and 7 I see that several smaller glaciers have not been mapped. Such random omissions need to be explained. A3: Several smaller glaciers have not been mapped in Fig. 5 and 7, and we explained it in the revised manuscript in lines 194-202.

Q4: As a minor point, please also improve the layout by inserting empty lines before section headings and before/after figures and tables if a re-submission is considered. A4: We have revised the manuscript according to the Referee's comments in the revised manuscript. Many thanks again.

Best wishes

Qin Ji

Please also note the supplement to this comment:
https://tc.copernicus.org/preprints/tc-2019-297/tc-2019-297-AC5-supplement.pdf

---

## Author Comment (AC7) · 2 Jul 2020

Q1: Introduction: The introduction is rambling and not to the point. It is not necessary in a journal specialist in cryosphere to explain what cryosphere is, for ex., or to mix lots of concepts, eg mass balance and SLR estimates, then water resources and formation of the Himalaya.

A1: Many thanks for your comments. We also clarified the objectives in the revised manuscript and corrected grammar and wording by a native speaker. In addition, we edited the introduction and study area according to your suggestion.

Q2: Study area: again this is rambling. Glacier classification into "continental" and "temperature glaciers" is erroneous, and references are outdated. References about

climate data are also outdated. This is not a climate paper so such details are meaningless. Showing pictures of Everest in figure 1 does not bring anything to the paper.

A2: According to the advice of anonymous referee, we deleted the sentence "Glacier classification into "continental" and "temperature glaciers"". In order to indicate the distribution of glaciers in the southern slope and northern slope, we made details in climate. Showing pictures of Everest in figure is the purpose to intuitively demonstrate the highest peaks in the world.

Q3: Data and Methods: No new methods were used here, and the authors state that for some years they averages +/- 2 years. It makes it hard to know then what year the glacier outlines represent when for some areas various years of imagery were used. Reporting cloud cover for each scene is irrelevant since cloud cover can be outside the glaciers. It becomes clear only later on that the authors derived ne set of outlines per decade. In this case, why not use RGI which is also from 2000 and is a compilation? The authors would need to show the superiority of their approach over existing outlines, but this is not done in the paper.

A3: The purposes of this paper analyzed the glacier distribution and variation characteristics in the entire Himalayas, so the method of glacier boundaries extraction is not new. The RGI are based on the ASTER data and this paper we extracted glacier outlines used Landsat TM/ETM+/OLI in 1990, 2005 and 2015. In order to avoid the impact of data inconsistency, we obtained the glacier boundaries used Landsat data and we also corrected potential error by visual interpretation in our paper.

Q4: Glacier mapping also is based on established methods (band ratios) and this section does not need almost one page to describe. Just a technical detail, band ratio of 1 seems very low for OLI, while 1.8 for TM seems acceptable; the authors do not comment on this. Also, the authors refer to the 2nd Chinese inventory but do not add a reference, nor how it was used. Debris covered glaciers, as it seems, were mapped manually but the description is fuzzy and there is nothing innovative here.

A4: Thanks for the Referee's suggestion we have added some contents in the details of image thresholding for Landsat OLI imagery and marked them in red in lines 183-190. In order to make the Landsat OLI data match with the Landsat TM/ETM+ imagery, we set Landsat TM/ETM+ as a reference to make geometric correction to Landsat OLI scenes. We also add the reference for 2nd Chinese inventory in line 204 in the revised manuscript.

Q5: Error estimates are based only on a single glacier and seem incomplete. A study of such spatial extent would need a much more thorough error and uncertainty section.

A5: Many thanks for your advice and we have used the buffer method to calculate the accuracy in lines 245-247 in the manuscript.

Q6: The authors present glacier changes across the study area with respect to various factors: part of the range, elevation bin, type of glaciers etc. While this is of possible interest, especially with respect to the spatial distribution of glacier changes, the information presented is hard to distill and very dense. This needs much more synthesis. Also, the changes are referred to most of the times as simply "changes"- it should be mentioned when the changes are in glacier area, length or height etc.

A6: The "changes" in this paper is referred to the area change.

Q7: Some concepts need much more development, for example debris cover. For example l 492 - 495 the authors mention the melt inhibition due to thin debris- however in the recent years there have been a number of publications which point at the presence of supra-glacial lakes and ice cliffs and their effect on melting rates over thick debris. The references presented are also outdated. Also the authors claim they test the effect of debris on melting rates- "To investigate whether the debris of the Himalaya can inhibit the glacier melting" (l522) - but then they present area changes, while debris cover glaciers can get thinner yet display no area change. So area change as an indicator of surface melt is not an appropriate measure.

A7: The Referee's comments are good. However, the large area covered by glaciers in the Himalayas, it is more difficult to calculate the change in height, so we used area change as an indirectly indicator.

Q8: In general results are difficult to follow in the form they are presented. For example, the authors compare the glacier change analysis with other studies- but there is no reference as to which year, so this is quite meaningless without adding more details. For example, they show only 1.3

A8: Many thanks for the Referee's advice. We wanted to analysis the relationship between glaciers and topography, so we statistics the area in different elevation zones of the Himalayas mountain.

We appreciate for the Referee's warm work earnestly, and hope that the corrections will meet with approval.

Best wishes

Qin Ji

Please also note the supplement to this comment:
https://tc.copernicus.org/preprints/tc-2019-297/tc-2019-297-AC7-supplement.pdf